# Personal and Social Responsibility Programme Effects, Prosocial Behaviours, and Physical Activity Levels in Adolescents and Their Families

**DOI:** 10.3390/ijerph17093184

**Published:** 2020-05-03

**Authors:** Juana García-García, David Manzano-Sánchez, Noelia Belando-Pedreño, Alfonso Valero-Valenzuela

**Affiliations:** 1Health, Physical Activity and Education (SAFE) Research Group, Sport Sciences Faculty, University of Murcia, 30720 Santiago de la Ribera, Spain; garciagarcia.jn@gmail.com; 2Department of Physical Activity and Sport, CEI Campus Mare Nostrum, University of Murcia, 30720 Santiago de la Ribera, Spain; david.manzano@um.es; 3 Faculty of Sports Science, Universidad Europea de Madrid, 28670 Villaviciosa de Odón, Spain; 4Department of Physical Activity and Sport, Sport Sciences Faculty, CEI Campus Mare Nostrum, University of Murcia, 30720 Santiago de la Ribera, Spain; avalero@um.es

**Keywords:** responsibility, teaching methodology, values, physical education, prosocial behaviours

## Abstract

The aim of this study was to analyse a personal and social responsibility programme in students and their family’s perceptions relative to responsibility, prosocial behaviours, empathy, violence perception and physical activity levels. A sample consisting of 57 physical education students between 11 and 14 years old (mean (M) = 11.93; standard deviation (SD) = 0.73) that included 32 of their parents (M = 49.31; SD = 6.39) was distributed into experimental and control groups. The main results indicate that there were initial significant differences in favour of the control group for personal and social responsibility compared to the experimental group and they disappeared at the end of the treatment. There was an increase in antisocial behaviours for the control group at the end of the treatment. The experimental group also enhanced the values in violence perception for both students and families as compared to the control group. These results seem contradictory, which may be due in part to a short-time intervention programme and a low number of participants in the sample. More studies will clarify the improvements this kind of programme can bring to the variables studied.

## 1. Introduction

An important amount of the mortality and morbidity experienced around the world today can be prevented [1]. The major health determinants are socioeconomic and lifestyle factors, next to the physical environment. The European health report of the World Health Organization (WHO) included physical activity as one of the most important aspects in this lifestyle [2].

There is a large body of evidence demonstrating the beneficial effects of physical activity (PA) on physical and psychological health in youth [3]. However, only 19% of young people undertake PA at a level that meets these PA guidelines [4]. To address this problem, school physical education (PE) is an important setting to help youth to engage in PA at levels that contribute toward meeting current PA recommendations [5]. For that, PA can be understood as an essential resource for the holistic education of children and young people [6]. 

Schools are considered essential as places for education, interaction, and interdisciplinary learning, but the education system still overemphasises academic results compared to advances and progress in personal development [7]. Based on this educational need, the perception of education from the point of view of positive psychology [8,9] and emotional intelligence [10,11] is considered of special relevance, since they are knowledge paradigms focused on the development of personal wellbeing and psychological stability of people through the exploration, identification and emotions approach [6,12]. 

Nowadays, studies are looking for strategies to improve PA levels in schools, within the PE and after-school sports activity context, which has become the best place thanks to the variety of activities carried out and the relationships between teacher and students in different environments [13]. In this way, a suitable framework of the teaching-learning process can improve students’ responsibility and lead to a better development of social and emotional student skills [14,15]. Educational intervention programmes should promote socio-emotional skills with concrete, structured and targeted PA to improve the emotional intelligence [16,17].

In this sense, the Hellison Model of Personal and Social Responsibility (TPSR) [18,19] should be highlighted in the educational field, as beneficial effects on personal behaviour in young people have been observed. TPSR is a curriculum and instructional model that allows the promotion of different values, character, responsibility, and life skills in PE and after-school contexts [20,21,22,23]. In this sense, it has been field-tested for 40 years in several settings, predominantly in underserved urban environments [24]. In Sánchez-Alcaraz’s [25] systematic review, numerous benefits have been reported from the 35 studies which applied this methodology, and in all of them were found improvements. However, even though some studies have analysed teacher opinion with this methodology [26,27] no one has inquired of families about their children’s responsibility perception.

Specifically, TPSR has shown a positive relation with the intention of being physically active [28,29], and it is even used in PE classes to improve fair play and self-control [30] and after-school activities to both improve competences and useful skill for sports to develop educational values for students’ daily lives [31]. Following Escriva-Boulley [32], the methodology a teacher uses in his classes is key to improve the motivation and the amount of moderate-to-vigorous PA.

On the other hand, pro-social behaviours, understood as voluntary behaviours aimed at benefiting others, play an essential role in forming positive interpersonal relationships and maintaining well-being [33]. Their study in the school context is of great interest because of the relationship with intrinsic motivation and autonomy [34]. TPSR can be a model-based programme that allows this kind of behaviour in adolescents and children through its application in the school setting [14,23,35], decreasing antisocial behaviours in adolescents [36].

At another level, violence, defined as intentional behaviour that causes harm or prejudice [14] has been related to the classroom coexistence climate, values and pro-social behaviours [37,38]. Furthermore, responsibility has also been linked to perceived violence [39], with sportsmanship and personal and social responsibility negatively correlated with school violence and disruptive behaviour in physical education classes. Menéndez and Fernández-Río [21], confirmed the use of this model improved behaviour against violence with disruptive students in PE classes.

Finally, empathy is defined as an emotional response of understanding the emotional state of others, which induces a person to feel the state of those others [22]. Empathy, next to pro-social behaviours, is a predictor of personal and social responsibility in school children [40]. However, until now, no studies have been found that analyse the effect of TPSR programme in empathy. 

Therefore, the aim of this study is to analyse the effect that TPSR has on students and their parents’ perception of responsibility, pro-social behaviours and empathy (only in students), violence perception and PA level. Accordingly, it was hypothesised that students who experience the TPSR model will enhance their pro-social behaviours, empathy and violence perception and will have higher PA levels, with their families obtaining similar changes to their perceptions. 

## 2. Materials and Methods

### 2.1. Design

A quasi-experimental design study with a longitudinal quantitative methodology was carried out, with a pre-test and a post-test applied in a non-equivalent control group [41]. Educational centres were selected for accessibility and convenience [42]. The ethical criteria were fulfilled by obtaining a favourable report from the Bioethics Committee of the University of Murcia (registration number 630), which allowed this research to be carried out.

### 2.2. Participants

Students: 57 PE students participated, structured in an experimental group of 26 students (belonging to one classroom of 6th year of Primary and another classroom of 1st year of ESO (Educación Secundaria Obligatoria); 16 boys and 10 girls) and a control group of 31 students (belonging to one class of 6th year of Primary and another class of 1st year of ESO; 16 boys and 15 girls) from two public education centres in Murcia with similar characteristics and socio-demographic environments. Student ages were between 11–14 years (mean (M) = 11.93; standard deviation (SD) = 0.73). The participants corresponded to different Educational Centres and were predetermined by said Centres, implying the impossibility of their randomization. The inclusion criteria used to participate in the research were the following: similar socio-demographic characteristics, completion of all questionnaires and regular class attendance. In addition, none of the participants had previous experience with TPSR.

Parents: 32 parents (9 males and 23 females) participated, 16 of whom were in the experimental group and 16 in the control group. Their ages ranged from 32 to 57 years (M = 49.31; SD = 6.39). They were selected based on the completion of all the questionnaires for their child participating in the study. After the first meeting with the head and teachers of the educational centres, an informative meeting was held with the parents and researchers to explain to them personally the development of the project. Because the participating students were minors, the necessary permissions for the application of the methodology and the recording of the sessions were requested from the teachers and parents/guardians of the participating students. At this meeting, parents or legal guardians were encouraged to participate in the research. It was established as a necessary requirement that the same adult should fill in the questionnaires before and after the intervention. No gender preferences were established with respect to the participation of the parent or legal figure.

### 2.3. Measures

A questionnaire composed of the following scales was used for the analysis of the variables under study:

Personal and social responsibility. The Spanish validation [43] of the Personal and Social Responsibility Questionnaire (PSRQ) [39] was used to measure the personal and social responsibility of the participants (aged 9–15). The questionnaire consists of 14 items arranged in two factors of seven items each: social responsibility (items 1 to 7 inclusive) and personal responsibility (items 8 to 13 inclusive and 14). The participants answer the items on a 6-point Likert scale, graduated from 1 (Totally disagree) to 6 (Totally agree); where item 14, posed in the negative, would be evaluated with the scale in reverse. The internal consistency of the total scale, measured by Cronbach’s alpha coefficient, was 0.87 and 0.89 in the pre- and post-tests, respectively. The different subscales were 0.84 and 0.88 for social responsibility and 0.76 and 0.77 for personal responsibility.

Prosocial and Antisocial Behaviour. The Spanish version [44] of the Adolescent Social Skills Inventory (TISS) [45] was used. This evaluates the social competence of adolescents (12–17 years) in relationships with their peers. This scale consists of 40 items grouped into two factors: prosocial behaviour and antisocial behaviour. Participants answer the items on a 6-point Likert-type scale from 1 (Does not describe me at all) to 6 (Describes me completely). Cronbach’s α values between the pre and post tests were 0.85 and 0.91 for the total scale. For the subscales they were 0.84 and 0.91 for prosocial behaviour and 0.87 and 0.92 for antisocial behaviour.

Empathy. The Spanish version of the Interpersonal Reactivity Index (IRI) was used for young people aged 13 to 18 years [22]. This scale is composed of 28 items distributed in four factors of seven items each and measures the integral concept of empathy: Perspective Taking (PT), Fantasy (FS), Empathic Concern (EC) and Personal Discomfort (PD). Participants answer the items on a 5-point Likert-type scale from 1 (Doesn’t describe me well) to 5 (Describes me very well) with negative items scored on the reverse scale. It was decided to include only the items stated positively (19 items in total), thus obtaining scores of 0.85 pre and 0.86 post for the total scale, and, for the different subscales, 0.85 pre and 0.86 post for PT; 0.73 pre and 0.86 post for FS; 0.85 pre and 0.86 post for EC; and 0.85 pre and 0.86 post for PD.

Violence. The Daily School Violence Questionnaire (DSVQ), validated in Spanish, from the California School Climate and Safety Survey (CSCSS) [40] was used for second- and third-year obligatory secondary education students (mean ages 13.44 years and 14.49 years) [46]. It evaluates the perception of having suffered violence from peers or having observed it. This scale is composed of a total of 14 items distributed in two factors: violence suffered (items from 1 to 8 inclusive) and violence observed (items from 9 to 14 inclusive). The items are presented in a 5-point Likert-type response format from 1 (Never) to 5 (Always). Cronbach’s α values between pre and post takes were 0.90 and 0.91 for the full scale. For the subscales, they were 0.82 and 0.89 for the violence suffered and 0.93 and 0.92 for the violence observed.

Comparative perception of the level of physical activity performed. The Spanish version for adolescents aged 13–17 years [47] of the Comparative Physical Activity Scale [42] was used. It evaluates the perception of the amount of PA performed according to the level of activity in the environment. This scale consists of a single question: ‘Comparing your child with others of the same age and sex, how much PA does he or she get?’ This item is answered in a 5-point Likert answer format from 1 (much less) to 5 (much more). From this scale, it is not possible to analyse the reliability of the questionnaire because it is only one item, but it is a validated instrument.

Perception of the level of physical activity practice. The Spanish version [47] of the Physician-Based Assessment and Counseling for Exercise (PACE) questionnaire [47] was administered. Although it was initially designed for adults, it was later validated for young people aged 13–17 years, obtaining an acceptable correlation with the measurement of the amount of PA performed by accelerometers [40]. This scale is composed of two items that evaluate the amount of PA practiced for at least 60 min in the previous week (PACE1) and in a normal week (PACE2). Participants answer the items on an 8-point Likert scale, 0 (0 days) to 7 (7 days). Cronbach’s α values between the pre and post takes were 0.79 and 0.87, respectively.

### 2.4. Design and Procedures

Initially, there was a meeting with the directors and physical education teachers of the participating schools, during which they were informed of the duration of the research and its impact on the quality of the teachers and the education received by young people. Because the participating students were minors, the necessary permissions for the implementation of the methodology and recording of the sessions were requested from the teachers and parents/guardians of the participating students.

The teachers in the control groups used a conventional teaching methodology in which there was no intentional manipulation of the variables contemplated in the research. This methodology had a classic session structure and was differentiated into three parts (warm-up, main part and return to calm). However, the teachers in the experimental groups used the methodology based on the TPSR to promote a climate of responsibility within subjects (personal) and between subjects (among the students in the tasks posed in class). The contents of the study were the same for both the control and experimental groups in the educational curriculum.

#### 2.4.1. Training Teachers

The training of implementers is a crucial aspect that influences the success of such intervention programmes, improving both the fidelity of implementation and the effects generated on students [43,44,45,46,47,48].

Training started with a detailed and concise meeting with the participating teachers on how to carry out their training based on Donald Hellison’s Personal and Social Responsibility Model (TPSR), at which time they were given a small initial theoretical test to check their starting level on the model. From there and during the first weeks of the school year, the training of the teachers on the methodology to be implemented in the experimental groups began, with a total duration of approximately 30 h. To this end, they attended different training seminars in which the necessary tools were explained and provided to modify the method of teaching the contents of PE according to the TPSR [49], using the pedagogical strategies and curricular materials according to the characteristics and needs of their educational centres. In addition, in order to promote their training, they were given a manual called ‘Quick Guide to Hellison’s Methodology’ that brought together all the key aspects to be followed (levels and strategies, structure and application of the session, and strategies for conflict resolution). In one of these seminars, teachers were even given posters that reflected in a concrete way the different levels of the TPSR and were to be placed in visible places in the pavilion or classroom.

In addition, to guarantee treatment fidelity during the intervention, continuous training was provided to the teachers of the experimental group [50,51]. Face-to-face meetings between the principal investigator and teachers were held at a frequency of three weeks to check the extent to which the model was or was not being applied. In addition, there was full daily availability of the principal investigator via telephone or email to allow for follow-up and support. The objective was to continue with the training of teachers and to ensure the correct implementation of the TPSR.

#### 2.4.2. Fidelity Implementation

The teaching interventions were recorded for every sixth session of PE using a video camera (a total of five sessions was recorded, resulting in the recording of a total of 29 sessions with a duration of 55 min and a frequency of two sessions per week distributed for five months). A SONY HDR (High Dynamic Range) video camera (Tokyo, Japan) was placed on a tripod in one of the corners of the pavilion, so that the whole space where the sessions were held could be recorded; in addition, shadows or flashes were taken into account when placing the camera so that the quality of the image did not deteriorate. The camera, tripod and microphone worn by the teacher were present in the classes at all times to prevent them from influencing student behaviour. The camera was also in place during the six sessions prior to the start of the methodological intervention to allow the students to become familiar with it and accustomed to its presence so that they would not alter their behaviour when it was actually used in the process of applying the TPSR, thus favouring the appearance of spontaneous behaviour [52]. Observational criteria were determined to assess the validity of the intervention, such as not considering periods of non-observability, and a break in continuity of observation for periods of time greater than 10% of the total observation was not exceeded [53].

Observation was used as a means of collecting information through the tool for assessing responsibility-based education (TARE) instrument [48,54,55], originally created by Wright and Craig [56], which used a system of categories: (1) Model of respect (M); (2) Set expectations (E); (3) Provide opportunities for success (S); (4) Encourage social interaction (SI); (5) Assign tasks (T); (6) Leadership (L); (7) Grant choice and voice (V); (8) Role in assessment (A); and (9) Transfer (Tr). Additionally, when teachers finished a filmed session, they had to fill in a self-evaluation questionnaire that grouped the TARE categories in order to promote a reflection on the methodology implemented.

After the recording of each session, observational analysis was carried out, which necessitated the training of the observers. The principal researcher instructed them to make a proper record of the teacher’s performance in relation to the dimensions described above. Following Wright and Craig [56], the training began with the explanation and clarification of the TARE categories. Together with the principal investigator, two ‘example classes’ of PSR application using TARE were then observed. After the analysis, the observers shared the results obtained to pool criteria. Finally, the interobserver reliability (>88%) allowed the start of the analysis of the sessions; this was calculated with the following formula: AT = TA/A + D (total agreement = AT; total agreements = TA; agreements = A; disagreements = D) [56].

The analysis of all sessions revealed differences between the sessions analysed in the control group and the experimental group, with a percentage of more than 70% in all the categories analysed, except for the transfer section, which was 42%.

After receiving informed consent, the questionnaires for the control and the experimental group were passed out, under the supervision of the principal investigator and in the presence of the PE teacher or tutor. The teacher assisted in maintaining the anonymity and sincerity of the answers. Meanwhile, the principal investigator was attentive to resolving those doubts that were generated during the filling-in process. The environment in which the questionnaires were filled in was calm and quiet, in order to favour student relaxation and concentration during their individual completion.

In addition, on that same day, the teachers were given the booklets containing the questionnaires that they had to provide to the parents/guardians of the respective students, one per student. The same father, mother or guardian of the minor who completed the questionnaire in the pre-test would have to complete it in the post-test.

#### 2.4.3. Intervention

Personal and Social Responsibility Model (TPSR). The intervention began with the application of the TPSR in PE classes in the experimental group, teaching students the values of responsibility little by little and moving through the different levels [43]. This does not mean that previous levels cannot be reinforced, but rather that teachers and researchers must cooperate to establish the work plan that develops the priority levels and objectives. The physical education teacher taught the same content to both the control and experimental groups, although they adapted the content of their different teaching units to the session structure established during the training course to be used in the experimental group, composed of four parts based on the TPSR. 

In order to ensure that students from different classrooms in the experimental group received the same instructional programme, the use of strategies to promote responsibility were measured by the TARE. Table 1 shows the results in percentages of occurrence. The Mann–Whitney U test showed no significant differences in the teachers scores which denotes the use of similar methodology strategies between.

Conventional teaching. This is based on the development of the content and skills of the subject and on teacher-centred decision making [57]. The session format was composed of three parts: (a) warm-up in which the students prepared for the main part of the session and the tasks that were scheduled with joint mobility exercises and chase games; (b) main part in which the students performed a predesigned set of tasks to improve the selected skills (i.e., baseball batting drills, badminton hitting drills and games); and (c) back to calm in which the students performed lighter tasks to get ready for the next class (i.e., stretching exercises). The teacher decided when practice would start and stop. Students did not have to make decisions other than to participate in the different tasks. The teacher remained in full control of the class. To avoid possible biases in the study, participating teachers (supervised by the university research team) developed a lesson plan that could be appealing for the students (fun and enjoyable), but also of high quality for researchers and scholars. Tasks, drills, games and competitions were designed to increase students’ academic and active participation time [57].

Finally, after finishing the treatment, which coincided with the end of the school year, students and parents were given a post-test that repeated the same tests of the pre-test, thus allowing the evaluation of the effects achieved on the participants.

### 2.5. Data Analysis

The IBM SPSS 22.0 () statistical package was used, initially validating the instrument by analysing the internal consistency of both the pre-test and post-test for each of the scales using Cronbach’s Alpha test for reliability. An exploratory analysis of the data was carried out through box-whisker diagrams and descriptive measurements, which detected that the results could differ according to age; this was then taken into account in the inferential analysis.

In a first analysis, a multivariate analysis of variance (MANOVA) of repeated measures was carried out on the variables obtained from the different questionnaires in which the intra-subject factor was called test (with two levels: pre-test and post-test) and the group (with two levels: control and experimental) was considered as an inter-subject factor and age as a covariant for students. The same procedure was carried out for families, in which the intra-subject factor was called test, group and educational level were considered as an inter-subject factor, and age was not a covariant. An analysis of the residues revealed a lack of compliance with the hypothesis of normality in some of the variables, so a second analysis was carried out using non-parametric tests. The results obtained with both procedures were very similar and the results of the non-parametric tests have not been included for brevity. Due to the small sample size, the magnitude of change was obtained through the eta-squared effect size, which could be considered small (η^2^ = 0.01), medium (η^2^ = 0.06), or large (η^2^ = 0.14) [58,59]. 

## 3. Results

The means and standard deviations of the different variables in the pre-test and in the post-test, differentiating by group, are reflected in Table 2. In addition, the specific *p*-values are included with the MANOVA of repeated measures, the effect size through eta-squared and media differences between pre-test and post-test.

The contrasts proposed at the multivariate level for the inter-subject factors showed significant differences at the group level (Wilks’Λ = 0.594, F_(12,43)_ = 2.45, *p* = 0.016) and age (Wilks’Λ = 0.585, F_(12,43)_ = 2.54, *p* = 0.012). In the intra-subject analysis, there were no significant differences between the different measures over time (Wilks’Λ = 0.741, F_(12,43)_ = 1.252, *p* = 0.282), and there were no significant interactions according to group (Wilks’Λ = 0.830, F_(12,43)_ = 0.732, *p* = 0.713) or age (Wilks’Λ = 0.747, F_(12,43)_ = 1.212, *p* = 0.306). 

Due to the existence of differences for the inter-subject factor, the univariate contrast analysis was carried out to check for which specific variables these differences occurred. Significant differences were obtained for the group factor at the level of social responsibility (F_(1,54)_ = 4.607, *p* = 0.036) and personal responsibility (F_(1,54)_ = 4.717, *p* = 0.034), empathic concern (F_(1,54)_ = 4.360, *p* = 0.042) and observed violence (F_(1,54)_ = 5.792, *p* = 0.020). 

Finally, to know at what time the differences between the groups occurred, we proceeded to calculate the estimates between the group factor and time, finding significant differences with respect to personal responsibility (*p* = 0.022) and differences very close to significance for social responsibility (*p* = 0.052) with a medium effect size (η^2^ = 0.068) in favour of the control group in the pre-test, and the violence observed showed significant differences in favour of the experimental group in the post-test (*p* = 0.016).

Although no differences were found in the intra-subject analysis, after calculating the estimated marginal means, it was found that the mean in the post-test for the control group in the antisocial behaviours variable was significantly higher than in the pre-test (*p* = 0.009).

Focusing attention on the families, we proceeded in a similar way with the different variables measured in the students (Table 3). The contrasts proposed at the multivariate level for the inter-subject factors did not show significant differences at either the group level (Wilks’Λ = 0.644, F_(6,24)_ = 2.21, *p* = 0.077) or educational level (Wilks’Λ = 0.622, F_(6,24)_ = 2.43, *p* = 0.056). Regarding the intra-subject analysis, there were no significant differences between the different measures over time (Wilks’Λ = 0.734, F_(6,24)_ = 1.45, *p* = 0.237), not according to interaction with group (Wilks’Λ = 0.801, F_(6,24)_ = 1.00, *p* = 0.450) or educational level (Wilks’Λ = 0.716, F_(6,24)_ = 1.59, *p* = 0.195).

Although no differences were found at the multivariate level, the analyses of variance (ANOVAs) showed significant differences in the violence suffered for the time factor (F_(1,29)_ = 4.421, *p* = 0.044), time and educational level interaction (F_(1,29)_ = 4.518, *p* = 0.042) and time and group (F_(1,29)_ = 5.912, *p* = 0.021). Regarding the inter-subject factor, the univariate contrast analysis showed significant differences for the group factor at the level of social responsibility (F_(1,29)_ = 5.695, *p* = 0.024) and suffered violence (F_(1,29)_ = 8.827, *p* = 0.006). 

Finally, to know at what time the differences between the groups occurred, we proceeded to calculate the estimates between the group factor and time, finding significant differences at the level of violence suffered in the post-test (*p* = 0.001) in favour of the experimental group and differences very close to significance for social responsibility both in the pre-test (*p* = 0.060) and in the post-test (*p* = 0.055) with a large effect size in favour of the control group.

## 4. Discussion

The purpose of this study was to analyse the effects of TPSR in young students and their families on responsibility, prosocial behaviours, empathy, perception of violence and level of PA. According to this hypothesis, it was only partially confirmed. Regarding the effects on students, changes have been observed in the variables of personal responsibility, antisocial behaviours and observed violence. Specifically, the experimental group that received the methodological proposal based on the TPSR started from lower levels of personal responsibility and this was matched at the end of the intervention. Other studies in which the TPSR has been applied have obtained findings along this line, although with much stronger results in the improvement of both personal and social responsibility [48,49,60,61,62,63]. On the other hand, this study lacks more clarifying results for this type of variable, which could be explained in part by an insufficient intervention as pointed out by Carbonell et al. [60], who carried out an intervention of similar duration and concluded that a quarter was not enough time to achieve the full learning of responsibility. However, there is also some evidences of improvement in the application of TPSR in personal and social responsibility and reduction of negative behaviours with interventions lasting less than 3 months [26]. Following these outcomes, Manzano’s study [64] applied TPSR and students improved their personal but not social responsibility, suggesting longer interventions would be required in order to see results in the rest of the variables. 

Regarding prosocial behaviours, the results did not provide any significant relevance in the experimental group compared to other studies on the application of TPSR in the variables analysed [34,65]. However, it has been possible to verify changes in the control group who showed a worsening in their prosocial behaviours by significantly increasing their antisocial behaviours.

On the other hand, in relation to the effects on violence, there was a commitment to the observed violence in the experimental group. These results are contrary to other studies that have reported a decrease in disruptive behaviours in different contexts [39,66].

With regard to the level of perception of PA, although the improvements were not statistically significant, a positive trend (delta scores) was detected after implementation in the variables of the experimental group; meanwhile, in the control group these scores were far lower. These data are in line with the values of the Gómez-Mármol study [23] in which a greater development of responsibility is linked in those students who practice more PA.

Finally, we turn our attention to the second of the objectives set for the perceptions of families for the variables of responsibility, violence and PA levels carried out by their children. We have found no studies that try to see the evolution of the behaviour of families and the similarity in perceptions with respect to that of their children when this methodology is applied. However, there is some research that exposes how the use of interventions in socio-emotional aspects can reduce violence, especially in adolescents [66], being essential the role that other adults play as parents and teachers to cut down on these behaviours [67]. In this study, talking about violence perception similar behaviour was found in students and their families. However, contrary to what might be expected, the results obtained on these variables indicate worse perceptions in the experimental group after the intervention, specifically for the violence suffered by parents. This is in line with the students in the experimental group who yield worse perception in observed violence. In this sense, there is no a clear explanation, and more studies are necessary to clarify the improvements this kind of programme can bring.

Regarding the limitations of the study, we found that the low number of participants in the study (students, parents and teachers) could have affected the low significance of some of the analysed variables; therefore, the results should be viewed as preliminary and exploratory.

On the other hand, a longer period of TPSR implementation (for example, a whole school year) could allow teachers more dedication to promote each of the levels of the TPSR and thus generate more stable behavioural patterns over time as well as determine the significance between the results of the pre-test and post-test. For example, approving the project before the beginning of the academic year and training a group of teachers as soon they start the term can give better results thanks to a broader participation and coordination among the teachers [23]. Future research along the same lines is necessary, assessing the adequacy of the TPSR within the educational context to improve aspects related to emotional management, positive psychology and/or levels of PA. In turn, the use of interviews and field diaries to assess the implementation could be of great interest in order to detect gaps and possible lines of improvement from a practical and real point of view [27].

## 5. Conclusions

The application of the TPSR had a positive effect on the value of personal responsibility and a negative effect on perceived violence, as well as a positive trend on the perception of PA carried out, while their peers in the control group showed an increase in antisocial behaviours.

Families’ perception of their children’s behaviours was very similar to that of the students themselves, with an increase in the perception of violence at the end of treatment, and an upward trend in variables related to physical activity.

## Figures and Tables

**Table 1 ijerph-17-03184-t001:** Teachers’ strategies used to promote responsibility.

The Teacher	Teacher 1	Teacher 2	Mann-Whitney *U* Test Value
1. is a model for respect (M)	96.00	92.72	0.635
2. sets expectations (E)	89.08	83.63	0.914
3. provides opportunities for success (S)	86.36	81.81	0.588
4. fosters social interaction (SI)	81.81	79.99	0.661
5. assigns tasks (T)	90.90	79.99	0.107
6. provides leadership opportunities (L)	81.81	78.17	0.663
7. concedes choice and voice (V)	79.99	85.45	0.319
8. sets evaluation roles (A)	70.90	74.54	0.667
9. fosters transference (Tr)	40.00	43.63	0.663

**Table 2 ijerph-17-03184-t002:** Analysis of intervention results (students).

	Group	Pre-Test		Post-Test		Pre-Post Difference	
		Mean	SD	Mean	SD	Delta Score	*p*-Value
Social Responsibility	Control	4.73	0.66	4.79	0.53	−0.06	0.487
	Experimental	4.33	0.62	4.42	0.84	−0.09	0.548
	*p*-value + η2	0.052	0.068	0.074	0.058		
Personal Responsibility	Control	5.33	0.84	5.37	0.75	−0.04	0.873
	Experimental	4.73	0.80	4.98	0.88	−0.25	0.094
	*p*-value + η2	0.022 *	0.093	0.197	0.031		
Prosocial behaviour	Control	4.32	0.89	4.36	1.16	−0.04	0.916
	Experimental	4.38	0.72	4.15	0.77	0.23	0.336
	*p*-value + η2	0.777	0.001	0.572	0.006		
Antisocial behaviour	Control	2.53	0.99	3.05	1.22	−0.52	0.009 **
	Experimental	2.57	0.82	2.76	0.98	−0.19	0.354
	*p*-value + η2	0.529	0.007	0.116	0.045		
Perspective Taking	Control	3.56	1.04	3.72	1.09	−0.23	0.567
	Experimental	3.31	0.83	3.24	0.76	−0.04	0.967
	*p*-value + η2	0.345	0.017	0.150	0.038		
Fantasy	Control	3.17	0.92	3.08	0.83	0.04	0.579
	Experimental	2.91	1.07	2.67	1.05	0.20	0.270
	*p*-value + η2	0.275	0.022	0.111	0.046		
Empathic Concern	Control	3.52	0.91	3.56	0.91	−0.13	0.932
	Experimental	3.10	0.92	3.13	0.76	−0.05	0.825
	*p*-value + η2	0.105	0.048	0.102	0.049		
Personal Discomfort	Control	2.80	0.58	2.87	0.57	−0.07	0.615
	Experimental	2.96	0.51	2.85	0.70	0.11	0.494
	*p*-value + η2	0.375	0.015	0.762	0.002		
Suffered Violence	Control	1.73	0.72	1.88	0.96	−0.15	0.124
	Experimental	1.80	0.67	1.83	0.80	−0.03	0.953
	*p*-value + η2	0.88	0.000	0.59	0.005		
Observed Violence	Control	2.40	1.04	2.32	1.07	0.08	0.634
	Experimental	3.11	1.23	3.17	1.14	−0.06	0.775
	*p*-value + η2	0.053	0.068	0.016 *	0.102		
Comparative PA level percepion	Control	3.13	1.14	3.19	1.16	−0.06	0.832
	Experimental	3.08	1.29	3.35	1.05	−0.27	0.281
	*p*-value + η2	0.981	0.000	0.420	0.012		
Perception of PA level	Control	3.44	1.84	3.50	1.81	−0.06	0.936
	Experimental	3.37	1.71	3.71	1.58	−0.34	0.282
	*p*-value + η2	0.850	0.001	0.561	0.006		

Note: * *p* < 0.05; ** *p* < 0.01; PA = Physical Activity; η2 = effect size; Delta Score = Difference of means.

**Table 3 ijerph-17-03184-t003:** Analysis of intervention results (parents).

	Group	Pre-Test		Post-Test		Pre-Post Difference	
		Mean	SD	Mean	SD	Delta Score	*p*-Value
Social Responsibility	Control	5.58	0.54	5.54	0.53	0.04	0.968
	Experimental	4.99	0.71	4.99	0.79	0.00	0.824
	*p*-value + η2	0.060	0.117	0.055	0.122		
Personal Responsibility	Control	5.22	0.80	5.27	0.58	−0.05	0.925
	Experimental	4.87	0.68	4.68	0.74	0.19	0.364
	*p*-value + η2	0.400	0.025	0.107	0.087		
Suffered Violence	Control	1.23	0.39	1.14	0.23	0.09	0.110
	Experimental	1.60	0.55	1.73	0.64	−0.13	0.059
	*p*-value + η2	0.086	0.098	0.001 **	0.323		
Observed Violence	Control	2.28	1.31	2.36	1.26	−0.08	0.931
	Experimental	3.15	0.61	3.22	1.04	−0.07	0.578
	*p*-value + η2	0.309	0.036	0.209	0.054		
Comparative PA level percepion	Control	3.38	1.09	3.62	1.15	−0.25	0.253
	Experimental	2.69	1.44	3.06	1.39	−0.37	0.463
	*p*-value + η2	0.350	0.030	0.235	0.048		
Perception of PA level	Control	3.28	1.59	3.91	1.89	−0.63	0.168
	Experimental	2.88	1.83	3.31	1.35	−0.43	0.342
	*p*-value + η2	0.825	0.002	0.591	0.010		

Note: ** *p* < 0.01; PA = Physical Activity; η2 = effect size; Delta Score = Difference of means.

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
