# Peer review of "Personal and Social Responsibility Programme Effects, Prosocial Behaviours, and Physical Activity Levels in Adolescents and Their Families"

_ijerph, 2020, doi:10.3390/ijerph17093184_

Round 1

Reviewer 1 Report

A manuscript of interest to the scientific community is presented. However, some issues need to be addressed for publication:

1) The abstract must be structured in the same paragraph.

2) Line 87. Put DT instead of SD.

3) The conclusions are few and should be more in-depth and linked to the objectives of the study.

Reviewer 2 Report

This is an interesting paper reporting results from an intervention study on Spanish students. The manuscript merits publications and it's relevant for the filed of Education in general, and Physical Education in particular. However, authors should deeply discuss the length of intervention duration, since it was mentioned as a limitation, how long an intervention might last and how it would be feasible across the school year? Since the instruments used were mainly quantitative, it would be relevant to also include qualitative information, such as the perception of enjoyment of the students involved. 

Reviewer 3 Report

The article entitled “Personal and Social Responsibility Programme effects, prosocial behaviours, and physical activity levels in adolescents and their families” aims/tackles a topic that could be interesting for the readers of the International Journal of Environmental Research and Public Health. The key points that should be pointed out are to develop an intervention study the effects of TPSR in young students and their families on responsibility, prosocial behaviours, empathy, perception of violence and level of PA with measures in 2 moments, to use students and family same time and the possibilities to transfer the outcomes to practice.

On the contrary, the article shows shortcomings, that considering them like a whole, they could hamper the recommendation of the publication. The shortcomings focus on a small sample, with initial differences with the control group (this is permissible), and above all the insufficient time of intervention that has led to not having positive results either in relation to the students or in their families. In other words, the intervention has not worked.

These issues are difficult to resolve. The only thing I can think of is to try other statistical analyses, which then detail or unify the whole sample and make an analysis path with only the pre-test data to analyze the relationship between the variables. It is a pity that after such a costly intervention the authors have not obtained good results. It is clear that this program (TPSR), when not developed over a longer period of time does not work. I have reviewed papers with the same problem, in which a short period of application leads to not yet seeing the benefits and the authors themselves humbly acknowledge this. Another problem that sometimes makes it not work well is the lack of knowledge of the technique of the teachers who apply it, but in favour of the authors I have to say, that I believe that the programme has been well planned, in that aspect. Although obviously checking whether the teachers have made mistakes in its application is complicated. In any case, below I detail all these aspects in a more specific way:

From my point of view, the title is correct.

In relation to the summary, there is a 0 before the point in SD, you don't put 0 when it can't be more than 1, in this case, it can be more than 1, so it seems that the number has been lost. Then in the age of the families, you do include it correctly. I think it is better to structure the abstract in one paragraph.

In the introduction, it is necessary to justify and support the constructed hypothesis, justifying the effects of the SRPT on the different variables to be worked on. It must be improved, giving coherence to the information, starting by justifying the importance of our adolescents acquiring certain personal and social responsibility skills, and acquiring healthy lifestyle habits... to gradually explain each of the variables studied in the article. In addition to the fact that the introduction is not a group of unconnected paragraphs, it occurs, for example, with the paragraph from line 44 to 47, when talking about Athletes that makes the reader lose sight of what the topic and the context of work really is, which is the school one. I would eliminate that paragraph, and give more cohesion to the rest.

The aim is correct, and the hypothesis must be correctly supported by what is stated in the introduction, with a good theoretical basis to confirm it.

As for the method, I believe that this is the part where the problems are to be found. As for the sample, it is well described, but it is small.

Since I started reading it, I was concerned about the duration of the programme, only 5 months, it is clear that the application of TPSR does not work if it is not extended over a longer period of time. It has happened in several previous studies and often does not offer good results in such a short period. It is true that I believe that the authors have only failed in this aspect of the planning, since the rest has been explained and planned correctly, which is a shame.

You point out that there are two experimental groups, well, I understand that you have added two classes, perhaps that should be eliminated, and say only that it is a group of 26 students, even if it is a sample of two classes. They have worked with several teachers, perhaps that aspect should also be controlled to avoid differences between them and see if they have influenced the results by controlling the effect of the intervention of each one.

32 parents were included in the study, what criteria were followed? is it not one member per family? is it both? Then better fathers and mothers and better clarify this aspect how the sample selection was developed and the number of fathers and mothers who participated.

We found in the sample children of 11 years old, who are of primary school with those of 14 years old who are of secondary school, clarify that they are of secondary school, and why there are children of 11 years old, if normally 11 and 12 years old are still of primary school.

It is appreciated that they include the ethics committee that approved the study since in addition to working with minors, they were recorded, but I think they should also include the registration number.

Regarding data analysis and results:

I can't understand why they use parametric tests with those subjects and no homogeneity of variances, let alone a multivariate test as repeated measures that require homogeneity.

In addition, with respect to the comparison with the control group, given that there are starting differences in post-test, you should use the delta scores with the t-test or its non-parametric equivalent. Perhaps with the delta scores, we could have different results and save the study, I don't know if the authors have proved it.

In relation to students and families there is practically no improvement in anything after the application of the program, or they are very small, even the effect sizes are small or unrelevant. According to Cohen, d values between 0.2 and 0.5 are understood as small, between 0.5 and 0.8 as medium and above 0.8 as large; rs between 0.1 and 0.3 as small, between 0.3 and 0.5 as medium and above 0.5 as large. They talk about moderate effect size with .26, that's debatable.

In the effect size you should use eta square, if you use repeated measures it is more adequate and you should report it also in the univariate contrast analysis of your tables.

The discussion and conclusion section talks about positive effects when there are none, I think we should reanalyze, using the delta scores, or doing a path analysis and see the interaction of the variables in the whole sample to take advantage of the data. And rewrite these sections. This is the only solution I can think of to save the study, in which it is clear that the authors have made a great effort, and that it has only been impoverished by the intervention time that has been scarce or insufficient and perhaps, the somewhat small sample.

If it were finally published, the section on limitations would have to include all the aspects that could have damaged the study.

I wish them much luck, it is a pity that the results have not been better with the great effort they have made, which is very evident.

Round 2

Reviewer 3 Report

Dear authors, I believe that you have made an enormous effort and you have considerably improved the quality of the paper.

Sometimes the results of our interventions are not as positive as we would like, but it is good that studies of this type are also published, to determine that these programs, require certain characteristics for their application to be effective, since they are very sensitive to foreign variables.

As far as I'm concerned, it only remains for me to congratulate you on the great work you have done.

They should only correct the 0 error in the standard deviation of the abstract, which I think they forgot to put in, even though they pointed out in the letter of change that they had done so.

Thank you very much.

Author Response

Dear Reviewer,

We are really pleased to hear from you these considerations. It is true, sometimes the results are different than we would like to obtain, but we also believe it is important inform about them, learn from our mistakes, give suggestions for future studies and offer to the scientific community the opportunity to improve what we have done.

Indeed, it has been an enormous effort but we are delighted to know you have a positive consideration.

Finally, you are totally right, we forgot write down the cero before the standard deviation dot value.

We have attached a file with the correction included in yellow color.

Yours faithfully,

Alfonso Valero.
